# Fibrilar Polymorphism of the Bacterial Extracellular Matrix Protein TasA

**DOI:** 10.3390/microorganisms9030529

**Published:** 2021-03-04

**Authors:** Mnar Ghrayeb, Shahar Hayet, Neta Lester-Zer, Yael Levi-Kalisman, Liraz Chai

**Affiliations:** 1Institute of Chemistry, The Hebrew University of Jerusalem, Edmond J. Safra Campus, Jerusalem 91904, Israel; mnar.ghrayeb@mail.huji.ac.il (M.G.); shahar.hayet@mail.huji.ac.il (S.H.); neta.lester@mail.huji.ac.il (N.L.-Z.); 2The Center for Nanoscience and Nanotechnology, The Hebrew University of Jerusalem, Edmond J. Safra Campus, Jerusalem 91904, Israel; yael.kalisman@mail.huji.ac.il; 3The Institute of Life Sciences, The Hebrew University of Jerusalem, Edmond J. Safra Campus, Jerusalem 91904, Israel

**Keywords:** functional amyloid, *Bacillus subtilis*, TasA, extracellular matrix, protein aggregation

## Abstract

Functional amyloid proteins often appear as fibers in extracellular matrices of microbial soft colonies. In contrast to disease-related amyloid structures, they serve a functional goal that benefits the organism that secretes them, which is the reason for the title “functional”. Biofilms are a specific example of a microbial community in which functional amyloid fibers play a role. Functional amyloid proteins contribute to the mechanical stability of biofilms and mediate the adhesion of the cells to themselves as well as to surfaces. Recently, it has been shown that functional amyloid proteins also play a regulatory role in biofilm development. TasA is the major proteinaceous fibrilar component of the extracellular matrix of biofilms made of the soil bacterium and Gram-positive *Bacillus subtilis*. We have previously shown, as later corroborated by others, that in acidic solutions, TasA forms compact aggregates that are composed of tangled fibers. Here, we show that in a neutral pH and above a certain TasA concentration, the fibers of TasA are elongated and straight and that they bundle up in highly concentrated salt solutions. TasA fibers resemble the canonic amyloid morphology; however, these fibers also bear an interesting nm-scale periodicity along the fiber axis. At the molecular level, TasA fibers contain a twisted β-sheet structure, as indicated by circular dichroism measurements. Our study shows that the morphology of TasA fibers depends on the environmental conditions. Different fibrilar morphologies may be related with different functional roles in biofilms, ranging from granting biofilms with a mechanical support to acting as antibiotic agents.

## 1. Introduction

Biofilms are surface-associated aggregates of cells that are held together by an extracellular matrix (ECM), a “molecular glue” that embraces the cells and mediates their adsorbance to surfaces [1,2]. The ECM is composed of biopolymers: proteins, polysaccharides and nucleic acids that are secreted by the cells and assembled into a three-dimensional network [3]. Many of the ECM biopolymers form fibers that grant the biofilms with gel-like mechanical properties [4,5]. A specific example is that of functional amyloid proteins, fiber-forming proteins in the ECM of biofilms that share structural features with disease-related amyloid proteins. However, unlike disease-related amyloid proteins, functional amyloids are related with a function that benefits the organism, rather than leading to its malfunction and/or to the development of a disease. Functional amyloid proteins appear in a fibrilar form in the ECM of biofilms of various species, with the most-studied amyloid proteins being Curli in *Escherichia coli* [6,7], FapC in *Pseudomonas aeruginosa* [8], phenol-soluble modulin (PSM) in *Staphylococcus aureous* [9,10], and TasA/TapA in *Bacillus subtilis* [11,12,13].

In their fibrilar form, amyloid proteins appear as µm-long and nm-thick fibers, and they share a cross-β-sheet structure at the molecular level. However, their morphology on the fibrilar level may differ, and they appear as straight fibers, ribbons, twisted or coiled ribbons [14]. Interestingly, in many cases, a single protein aggregates into multiple fibrilar morphologies, depending on the environmental conditions [15,16,17,18,19,20]. Examples of amyloid proteins that appear in multiple forms of fibers include Tau [14,21], α-synuclein [16,22], Aβ [19,22,23], human amylin peptides [24,25] and Transthyretin (TTR) [26]. Interestingly, fibers with different morphologies have been related with different levels of amyloid toxicity [20,27].

TasA is a major proteinaceous component in the ECM of *B. subtilis* biofilms [12]. In vitro, TasA appears in a variety of forms: monomers, oligomers and fibers [28,29]. In contrast to many other amyloid proteins that are disordered prior to aggregation [30], TasA is structured as a monomer/oligomer in solution [13,28,29], and therefore an external trigger is necessary to initiate its aggregation. We have previously shown that TasA forms fibers in an acidic environment. These fibers were tangled (low persistence length) and compact, and they did not resemble the straight fibers commonly formed by amyloid proteins. Here, we observed the aggregation of TasA at a neutral pH. Above a certain TasA concentration, elongated and straight fibers formed. In concentrated TasA solutions with highly concentrated salt solutions, the elongated and straight fibers of TasA bundled up. In addition, these fibers also encompassed an interesting nm-scale periodicity along the fibers axis. On a molecular level, TasA fibers exhibited a twisted β-sheet structure, as indicated by circular dichroism measurements.

TasA has multiple roles in the biofilms. The protein (or its aggregates) has antibacterial properties [31], and it plays a regulatory role in biofilm development [32,33]. In a fibrilar form, TasA acts as a scaffold, providing a wrinkled architecture to biofilms [12,34]. Transitioning between the different roles of TasA may be achieved by its different polymorphs. This study opens the way to an understanding of the relationships between the molecular and macroscopic structures of TasA fibers, as well as to correlating fibrilar structures with their function in biofilms.

## 2. Materials and Methods

### 2.1. Materials

Unless otherwise specified, all materials were purchased from Sigma-Aldrich (St. Louis, MO, USA).

### 2.2. Native TasA Purification

TasA was purified as described previously [12,28]. Briefly, *B. subtilis* (double mutant strain sinR eps) liquid cultures were grown in MSgg broth at a 1:100 dilution from an LB starter culture, under shaking conditions (225 rpm). After 16 h of growth at 37 °C, the cells were pelleted (10,000 g, 15 min, 4 °C) and extracted once with a saline extraction buffer [12]. The sample was probe sonicated (1518 J, 5 s pulse, and 2 s pulse off), and the supernatant was then collected after centrifugation (10,000 g, 15 min, 4 °C) and filtered through a 0.45 μm polyethersulfone (PES) bottle-top filter. The filtered supernatant was collected after centrifugation (17,000 g, 15 min, 4 °C), concentrated with Amicon centrifugal filter tubes and either incubated in saline extraction buffer (preparation method 1, shown below) or passed through a HiLoad 26/60 Superdex S200 sizing column that was pre-equilibrated with a 50 mM NaCl, 20 mM tris solution at pH 8 (preparation method 2, shown below). Unless otherwise specified in the text, we used 20 mM tris buffer solution and a pH of 8 and varied the NaCl concentration. Purified protein was lyophilized or stored in the buffer solution at −20 °C until used. The concentration of TasA was determined using a BCA protein assay (ThermoFisher Scientific, Waltham, MA, USA).

### 2.3. Atomic Force Microscopy

Atomic force microscopy (AFM) topography scans were performed using a BioScope Resolve™ BioAFM (Bruker, Santa Barbara, CA, USA). Height sensor AFM images were measured in Peak Force Quantitative Nanomechanical Mapping (QNM)mode using ScanAsyst-Air probes (Bruker, resonance frequency 70 kHz, spring constant 0.4 N/m, tip radius (nominal 2 nm). Cantilever spring constants were calibrated using the Thermal Tune noise method. TasA adsorption on mica was performed as follows: 20 µL of TasA protein was placed on a freshly cleaved mica and kept for 30 min in a humid atmosphere. The mica piece was then washed twice in water for 10 min and dried with a stream of Nitrogen. TasA was adsorbed from a 100 µg/mL protein in 20 mM tris, 50 mM NaCl pH 8.0. TasA fibers were prepared from a 200 µg/mL TasA solution in 20 mM tris, 1.5 M NaCl that was pre-incubated for 8 days at 4 °C (preparation method 2, shown below). Particle size was measured from AFM images using the cross-section tool in the NanoScope Analysis software. The diameter of TasA oligomers was measured with at least 151 different particles and at least 30 different particles for TasA fibers.

### 2.4. Circular Dichroism

Circular dichroism (CD) spectra were recorded using a MOS-500 Biologic spectropolarimeter in the range of 195−260 nm with a 1 nm step, and a 0.5 s collection time per step at room temperature. CD was performed using 200 μL TasA solutions (an average of four independent measurements with TasA concentration being 88 μg/mL, 121 μg/mL, 100 μg/mL and 95 μg/mL) in 20 mM tris buffer, 50 mM NaCl and a pH of 8.0. TasA fibers were formed from TasA preparations (700 µg/mL) incubated in 1.5 M saline extraction buffer at 4 °C for 8 days (preparation method 1, shown below). Measurements were performed in 1 mm quartz cuvettes. CD spectra were buffer subtracted, and ellipticity was converted to mean residue ellipticity (*MRE*) units using the following equation: $MRE=\frac{m{}^{\circ}*Mw}{10*L*C*AA}$, where *Mw* is the molecular weight of the protein in g/mol, *L* is the cuvette path length in cm, *C* is the protein concentration in mg/mL and *AA* is the number of the amino acids in TasA (*AA* = 234).

Secondary structure analysis of TasA oligomers was performed by using the software BestSel [35,36] and Dichroweb (CDSSTR algorithm) [37,38].

### 2.5. Cryogenic Transmission Electron Microscopy (Cryo-TEM)

A 3 μL droplet of TasA solution was deposited on a glow-discharged TEM grid (300 mesh Cu Lacey substrate grid; Ted Pella). The excess liquid was blotted off with filter paper, and the specimen was rapidly plunged into liquid ethane pre-cooled with liquid nitrogen in a controlled environment using a Vitrobot Mark IV (FEI). The vitrified samples were transferred to a cryo-specimen holder (Gatan 626) and examined at −177 °C using a FEI Tecnai 12 G^2^ TWIN TEM instrument operated at 120 kV in low-dose mode. Grids were imaged at a few micrometers under focus to increase phase contrast. The images were recorded with a 4K × 4K FEI Eagle charged-couple device (CCD) camera.

TasA fibers that were scanned with cryo-TEM were prepared as follows. TasA were fibers formed by incubating a 200 µg/mL TasA in 20 mM tris, 0.05 M and 1.5 M NaCl at 4 °C for 8 days (preparation method 2, shown below). Bundles of fibers were formed from a 700 µg/mL TasA preparation that was kept in 1.5 M saline extraction buffer [12] for 10 days at 4 °C (preparation method 1 below). The sample was dialyzed against 20 mM tris, 50 mM NaCl solution prior to cryo-TEM imaging. TasA aggregates in acid formed following the pH adjustment of a TasA solution (200 µg/mL in 20 mM tris, 50 mM NaCl) to 2.5 using formic acid (preparation method 3, shown below).

### 2.6. Turbidity Measurements

Turbidity measurements were performed using a Synergy H1 Biotek plate reader. The optical density (OD) at 350 nm was recorded over time for the following solutions: a 100 µg/mL TasA solution in 20 mM tris, 50 mM NaCl immediately after adjusting the pH to 2.5 with formic acid (preparation method 3, shown below), 100 µg/mL or 800 µg/mL of TasA in 120 mM tris and 1.8 M NaCl (preparation method 2, shown below). The presented data are buffer-subtracted.

### 2.7. Thioflavin-T (ThT) Fluorescence

ThT fluorescence intensity was recorded at room temperature using a Synergy H1 Biotek plate reader with 445 nm excitation and 492 nm emission wavelengths. Reaction mixtures similar to those used for the turbidity measurements, supplemented with 10 µM ThT, were placed on a black, clear-bottomed plate.

### 2.8. Fiber Preparation

TasA fibers were prepared under the following conditions:

Preparation method 1: Natively purified TasA was incubated in saline extraction buffer (1.5 M NaCl, 5 mM potassium phosphate buffer, pH 7, 100 mM 3-morpholinopropane-1-sulfonic acid (MOPS), pH 7, 2 mM MgCl_2_) for a minimum of three days. The protein concentrations ranged between 0.7–3 mg/mL.

Preparation method 2: Natively purified TasA was passed through a HiLoad 26/60 Superdex S200 sizing column that was pre-equilibrated with a 50 mM NaCl, 20 mM tris solution at pH 8 and lyophilized. The protein was then dissolved with aqueous NaCl solutions to adjust the protein and salt concentrations to 0.2–3 mg/mL and 1.5–2 M, respectively. The protein solutions were incubated at 4 °C for a minimum of two days.

Preparation method 3: Natively purified TasA was passed through a HiLoad 26/60 Superdex S200 sizing column that was pre-equilibrated with a 50 mM NaCl and 20 mM tris solution at pH 8. TasA aggregates were formed from freshly purified TasA by adjusting the solution pH to 2.5 with formic acid. TasA concentrations were 0.05–0.1 mg/mL TasA(non-lyophilized) or 0.2 mg/mL (lyophilized). The solutions were incubated at room temperature for a minimum of three hours.

## 3. Results

### 3.1. TasA Forms Fibers with Different Morphologies Depending on the Environmental Conditions

We used cryo-TEM to study the morphology of TasA fibers that formed by the incubation of 200 µg/mL protein solutions at pH 8 for a minimum of two days at 4 °C (preparation method 2, described above). Fibers that formed at low salt concentrations (50 Mm NaCl) were straight, µm-long and nm-scale thick and reminiscent of canonic amyloid fibers (Figure 1a). Similar fibers formed in the presence of a highly concentrated salt solution (1.5 M NaCl) (Figure 1b). For comparison, we have added a cryo-TEM image of TasA fibers that were formed in an acidic solution of pH 2.5 in Figure 1c (preparation method 3). These fibers were significantly different than those that formed at neutral pH. They were tangled, thicker and had a persistence length that was much smaller than the straight fibers that formed at neutral pH and a high protein/salt concentration.

### 3.2. TasA Fibers Carry a Distinct Periodicity

We examined the TasA fibers that formed at a higher TasA concentration of 700 µg/mL (preparation method 2). At a low salt concentration (50 mM), the fibers that formed were similar to those that formed with a lower TasA concentration (200 µg/mL) (Appendix A). However, when both the protein and the salt concentrations were high, at 700 µg/mL TasA and 1.5 M NaCl (preparation method 1), bundles of thin fibers formed, as shown by cryo-TEM (Figure 2a). Similar fibril bundles formed using preparation method 2 with 2 mg/mL TasA at 2 M NaCl (Appendix A).

Observing the bundles of TasA with a higher magnification (Figure 2b) revealed that the fiber bundles exhibited a periodic structure, resembling beads on a string. The Fast Fourier transform (FFT) analysis of Figure 2b revealed that the pitch size along the bundles was 4.8 ± 0.5 nm, as indicated by the arcs appearing along the fiber axis in the FFT image (Figure 2b, inset). Interestingly, indications for similar fibrilar assemblies were observed by recombinant TasA preparations [13,39]. We also observed TasA fibers, similar to those observed in Figure 1b (200 µg/mL TasA in 1.5 M NaCl solution, preparation method 2), with a high-resolution AFM (Figure 2c). These fibers resembled the “beads on a string” morphology that was observed by cryo-TEM (Figure 2b). A cross-section along the fibers (marked with a white line in Figure 2c) showed that the pitch size was ~15 nm (Figure 2d). The discrepancy between the pitch and oligomer sizes measured with cryo-TEM and AFM may be attributed to tip convolution effects. The analysis of AFM images of TasA oligomers (see for example Figure 2e) yielded the size distribution of the oligomers (before they aggregate into fibers) that peaked at ~17 nm (Figure 2f). Interestingly, a similar distribution of sizes was measured in an analysis of the AFM images of the fibers, such as those presented in Figure 2c,g. The cryo-TEM and AFM images, together with the size distribution analysis, point at the possibility that the fibers formed via oligomer-to-oligomer addition, but we cannot rule out further rearrangements into twisted ribbons.

### 3.3. Fiber Morphology Depends on Aggregation Kinetics

We compared the aggregation kinetics of TasA in highly concentrated protein and salt solutions to the aggregation kinetics in an acidic solution using turbidity and thioflavin T (ThT) measurements. Turbidity measurements of 800 µg/mL TasA solutions in a 1.8 M salt solution (preparation method 2) show a slow aggregation process followed by a plateau at approximately 100 h (~4 days) (Figure 3, full black symbols). Using a lower TasA concentration (100 µg/mL) in 1.8 M NaCl solutions (preparation method 2), the turbidity measurements remained constant over time, indicating that TasA did not aggregate into fibers under these conditions (Figure 3, empty black symbols).

The inset to Figure 3 shows a turbidity measurement (at 350 nm) over time of TasA in an acidic solution of pH 2.5 (preparation method 3). Here, the protein aggregated quickly, in a time frame of around one hour, until the aggregates settled, as was indicated by a turbidity decrease. The slow aggregation of TasA in concentrated protein and salt solutions relative to its fast aggregation in acidic solutions suggests a different aggregation mechanism. We recently attributed the aggregation of TasA in acidic solutions to the changes in its surface potential and, after analyzing the aggregates’ morphology, we developed the analogy between TasA and colloidal aggregation [40]. In contrast, the slow fiber formation in concentrated salt solutions points at a nucleation–elongation mechanism [41,42].

The fibers that formed in salt reached higher plateau turbidity values relative to the peak turbidity values of the fibers formed in acid. This suggests that the aggregates that formed in acid were denser than those that formed in salt, and therefore they settled early in the aggregation process. This observation is in agreement with the finding that the morphology of the fibers formed in acid was more compact than the long and thin fibers formed in salt.

We performed a thioflavin T (ThT) assay in addition to the kinetic turbidity measurements in order to evaluate the dynamic structural changes of TasA in the presence of salt. Previously, we showed that the ThT fluorescence increased in response to the reduction of the pH in solution to 2.5, indicating the formation of β-sheet structures [43]. The ThT data of the aggregation of TasA in salt are colored red on the right Y axis in Figure 3, with the full and empty symbols corresponding to 800 µg/mL and 100 μg/mL TasA solutions, respectively. Both measurements were performed in 1.8 M NaCl solution (preparation method 2). The ThT measurements traced the turbidity measurements in both concentrations of TasA. Specifically, there was an increase in the ThT fluorescence in the concentrated TasA solutions over time (full red symbols), indicating the formation of β-sheet structures. Such an increase was not detected with the lower TasA concentrations (empty red symbols), suggesting that, under these conditions, when TasA did not aggregate into fibers, it also did not gain an amyloid-like β-sheet structure.

### 3.4. TasA Bundles Contain Twisted β Sheets

Following the ThT indications that the TasA fibers and bundles gained β-sheet structures during the aggregation in a concentrated salt solution, we were interested in measuring their secondary structures using circular dichroism (CD). Figure 4a shows the CD spectra of TasA oligomers prior to aggregation (average of four measurements, with 100 µg/mL average TasA concentration in 20 mM tris, 50 mM NaCl and pH 8.0). TasA oligomers were structured in solution, as has been previously shown [28,29]. Fitting the CD spectra of TasA oligomers with linear combinations of CD spectra of known protein data sets using the analysis software Dichroweb [37,38] and Bestsel [35,36] yielded similar secondary structure components of TasA oligomers: the helical percentage (~20%), β-sheet content (20–30%) as well as turns (~20%) and other undetermined structures (30–40%) (Table 1).

The CD spectrum of the TasA fibers (700 µg/mL) in a concentrated salt solution (1.5 M saline extraction buffer, preparation method 1), (Figure 4b, blue line), plotted together with the CD spectra of TasA oligomers (Figure 4b, black line), shows that increasing the protein and salt concentrations resulted in a significant structural change of TasA. Specifically, the spectrum has a dominant peak at ~223 nm, which is indicative of the formation of a twisted β-sheet structure [22,35,44]. We have previously shown that TasA settles as the pH changes to 2.5, and therefore its CD spectrum cannot be recorded [28,45].

## 4. Discussion

We have shown that native TasA aggregated into fibers at a neutral pH when the concentration of the protein was larger than (or equal to) 200 µg/mL. Under these conditions, TasA fibers were µm-long and straight, unlike the tangled and low-persistence length fibers that formed in acidic solutions, as shown previously [40]. In the presence of highly concentrated salt solutions and a high TasA concentration (700 µg/mL), these fibers bundled up. Figure 5 summarizes the conditions that lead to the formation of TasA fibers and correlates these environmental conditions with a schematic representation of the fibers’ morphology.

To account for the different morphologies of TasA in the different environments, we suggest the following aggregation mechanism. We previously showed that native TasA was isolated in the form of structured oligomers in a neutral pH and low salt concentration [28]. This was the first study that pointed at the polymorphism of TasA at the molecular level, based on CD results. The CD spectrum of TasA oligomers shows two peaks at ~208 nm and 222 nm, which are typical for an α-helical secondary structure. A software analysis of the CD data indicates that, in addition to the α helical content, TasA oligomers also contain a fair percentage of β sheets (Figure 4a, and Table 1). Indeed, recent findings regarding the crystal and nuclear magnetic resonance (NMR) structures of TasA monomers have shown that TasA consists of a jellyroll fold composed of antiparallel β-sheet, short helices and loop regions [29]. The CD spectrum of TasA oligomers resembles the calculated CD spectrum of antiparallel β structures [35], but its two minima are somewhat shifted at ~208 nm and 225 nm [28,29]. Given the crystal structure of TasA [29], this CD spectrum may reflect a combination of β-sheets as well as α helices in the structure of TasA in solution (when both the protein and the salt concentrations are low).

In an acidic environment, abrupt changes in surface charge induced the aggregation of the oligomers into compact aggregates. However, the fibers formed from TasA of a similar concentration in neutral solutions were dramatically different in morphology (Figure 1). This difference points at a different aggregation pathway in acid and at a neutral pH. Diehl et al. used NMR and X-ray diffraction and showed that the structure of aggregated TasA, formed in acidic solutions, is the canonic cross-β-sheet [29]. Since the aggregation is abrupt, we expect structural rearrangements to occur post-aggregation. In a solution of concentrated TasA concentration (>200 µg/mL), oligomer–oligomer collisions may be promoted. The aggregation into fibers is very slow under these conditions, indicating that nuclei need to form prior to fiber elongation. During the nucleation period, structural changes may occur within the oligomers of TasA, which could induce the addition of oligomers to the nuclei and eventually to fiber elongation. In the presence of highly concentrated salt solutions, the salt screens the surface charge of the fibrils and allows the formation of bundles.

The CD spectrum of TasA bundles shows a negative peak at ~223 nm (Figure 4b), which is shifted relative to the canonic β-sheet peak at 215 nm. Such a peak shift corresponds with twisted β sheets, as has been predicted by calculations [35] and was observed also with c-terminal truncated α-synuclein [22]. We therefore speculate that β-sheet twisting occurred during the nucleation stage and that it caused neighboring oligomers and/or nuclei to stick and fibers to elongate. This is in line with an NMR study of TasA fibrils by El Mammeri et al., who suggested that, upon aggregation, a TasA core remains globular, while the β-strands rearrange into a cross-β-sheet assembly [13].

In the next step, oligomers or deformed oligomers were packed via intermolecular interactions. Differences in the structure of the oligomers and/or the different packing of the oligomers may lead to differences in fiber morphology. For example, it has been recently shown that α-synuclein formed distinct polymorph fibrils even though they shared a common kernel [19]. The differences in the morphology of the fibers on a larger scale were attributed to the different packing of the protein kernel. Borrowing these ideas for the formation of TasA fibers, we suggest that the oligomers may act as a kernel—a basic repeating unit in the fibers. Molecular structural changes in the oligomers and/or differences in their packing during the formation of the fibers may account for the polymorphism at the fibril level.

## 5. Conclusions

Our study focused on the fibril-level assembly of TasA oligomers. TasA oligomers aggregate into fibers of different morphologies, depending on the environmental conditions. Different polymorphs of TasA convey the exposure of different functional groups at the surface of the fibers and they may also impose differences on their mechanical properties. Our study suggests that controlling the pH, salt and protein concentration in solution may be used in biofilms as a means to control the fibrilar morphology and thereby the fate of the fibers of TasA. Which environmental conditions actually reside in the proximity of the cells in biofilms and how they affect the morphology of TasA fibers at the molecular and global fibrilar levels are yet to be determined.

## Figures and Tables

**Figure 1 microorganisms-09-00529-f001:**
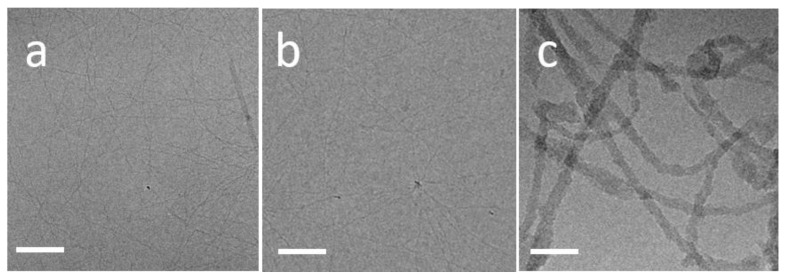
TasA fibrilar polymorphism. Cryo-TEM images of TasA fibers that formed in different environments. Straight fibers formed at 200 µg/mL protein at (**a**) 50 mM NaCl and (**b**) 1.5 M NaCl. (**c**) TasA aggregates that formed from a 200 µg/mL protein solution, 50 mM NaCl, after adjusting the pH to 2.5 were composed of tangled fibers. Scale bars correspond to 100 nm.

**Figure 2 microorganisms-09-00529-f002:**
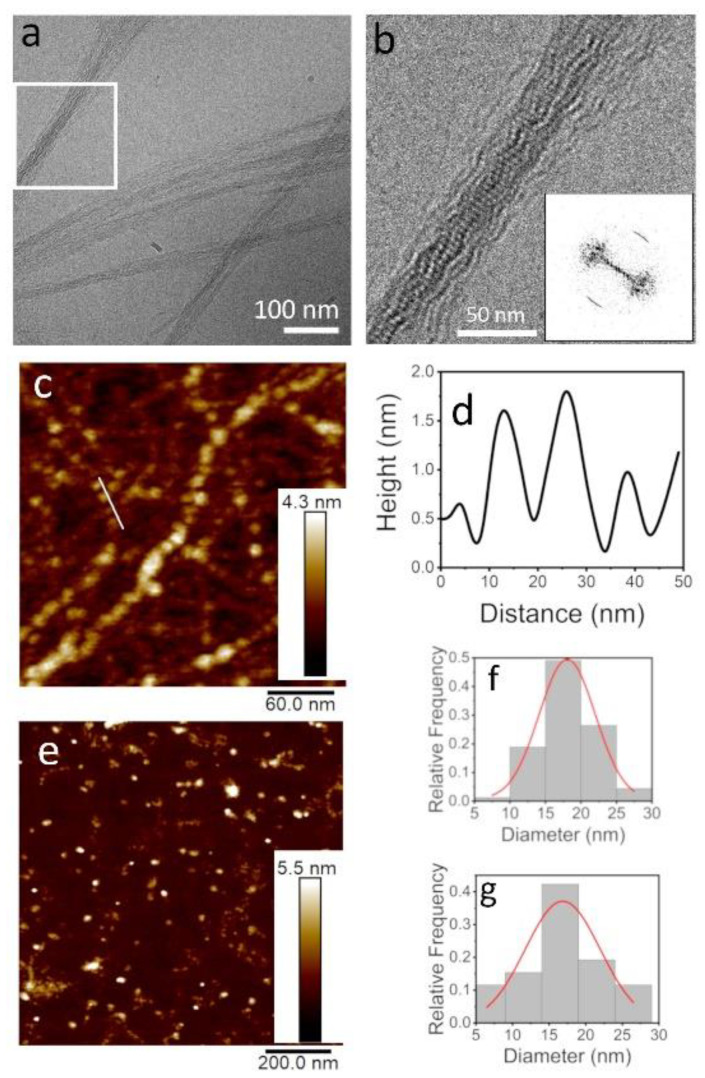
TasA bundles are composed of globules and exhibit a distinct order along the fiber axis. Cryo-TEM image of TasA bundles (**a**) together with a zoom into the squared area in (**b**) show that TasA bundles are composed of an ordered array of globules. The Fast Fourier transform (FFT) analysis of image b marks the pitch size along the fiber axis as 4.8 ± 0.5 nm (b, inset). The atomic force microscope (AFM) view of TasA fibers (200 µg/mL in 1.5 M NaCl) shows that separate fibers (not in a bundle) are also composed of globules (**c**). Height scale bar appears to the right of the AFM images. A cross-section along a fiber (marked with a white line in c) shows a periodic structure of globules with a ~15 nm pitch (**d**). The AFM image in (**e**) shows separate oligomers (100 µg/mL, 20 mM tris 50 mM NaCl, pH 8.0) on a mica surface, prior to aggregation. The analysis of the size distribution of both the TasA oligomers (**f**) and the globules in the fibers (**g**) yielded a diameter of ~17 nm. The discrepancy between the AFM and TEM dimensions is explained in the text.

**Figure 3 microorganisms-09-00529-f003:**
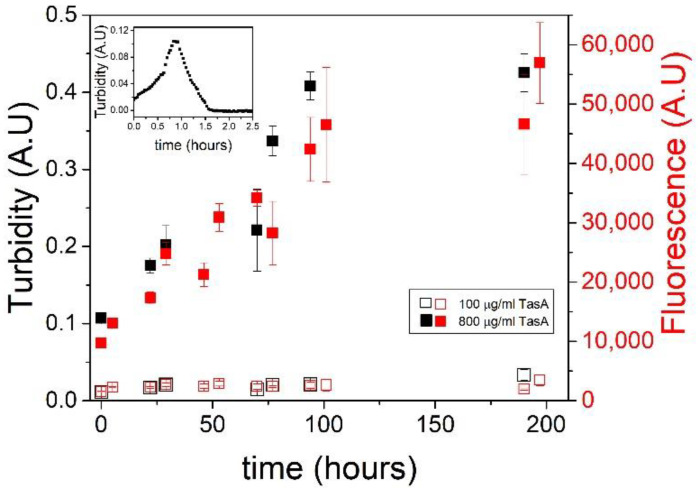
TasA exhibits different aggregation kinetics in acid and in salt solutions. Turbidity measurements of TasA aggregation in a salt solution (1.8 M NaCl, 120 mM tris, pH 8.0) are shown in the main figure, and turbidity measurements in an acidic solution are shown in the inset. Two different TasA concentration were used. TasA aggregated at high protein concentration (800 µg/mL protein, full black symbols) but not at low concentration (100 µg/mL, empty black symbols). Thioflavin T fluorescence assays of the aggregation of TasA in salt (1.8 M NaCl, 120 mM tris, pH 8.0) show an increase of fluorescence over time (800 µg/mL TasA, full red symbols), indicating the formation of β sheet structures. At a low TasA concentration (100 µg/mL), aggregation was negligible (empty red symbols), as indicated by the low ThT fluorescence under these conditions. In an acidic solution (inset), TasA aggregation (100 µg/mL TasA, 20 mM tris, 50 mM NaCl, pH 2.5) was faster than in salt, and the aggregates settled after ~1 h.

**Figure 4 microorganisms-09-00529-f004:**
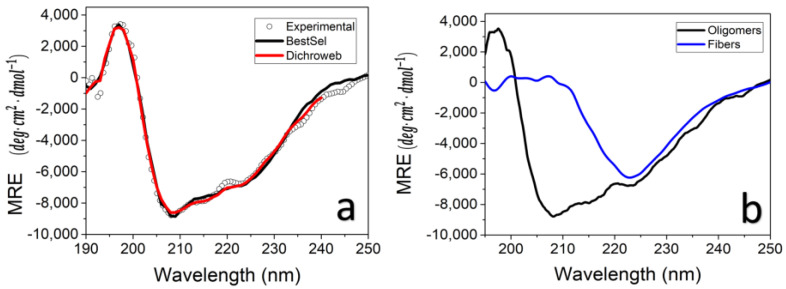
TasA changes structure during fiber formation. Circular dichroism (CD) spectrum of TasA oligomers, (100 µg/mL average concentration in 20 mM tris, 50 mM NaCl, pH 8) (empty symbols) and best fits to the data using the CD analysis software Dichroweb (red line) and BestSel (black line) (**a**). The CD spectrum of TasA oligomers (black line) is plotted together with that of TasA in the form of fibers (700 μg/mL, 1.5 M NaCl) following an eight-day incubation at 4 °C in saline extraction buffer (blue line) (**b**). We attribute the change in the CD spectrum to twisting of the β sheets upon aggregation into fibers [35].

**Figure 5 microorganisms-09-00529-f005:**
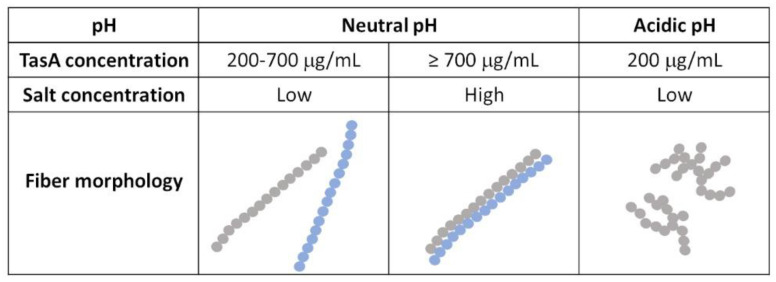
Summary of the conditions used in this study and the corresponding fiber morphology of the TasA fibers (schematic representation). Low salt concentration is 50 Mm, high salt concentration is 1.5 M–2 M.

**Table 1 microorganisms-09-00529-t001:** Analysis of the CD spectra obtained for TasA oligomers (Figure 4a) using the software BestSel [35,36] and Dichroweb [37,38]. The table specifies the percentage of dominant secondary structures in TasA oligomers.

	Helix	β Sheet	Turn	Others
**BestSel**	18.6	21.6	16.6	43.2
**Dichroweb**	19	28	23	29

## Data Availability

The data presented in this study are contained within the article and Appendix A.

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
