# Peer review of "Fibrilar Polymorphism of the Bacterial Extracellular Matrix Protein TasA"

_microorganisms, 2021, doi:10.3390/microorganisms9030529_

Round 1

Reviewer 1 Report

The presented article is dedicated to the investigation of the environmental factors affecting the TasA fibril morphology. The idea that different aggregates morphology may be adaptive in a different environment is attractive and need to be investigated. 
Unfortunately from my point of view, the authors did not find an answer for the mentioned auestion at the moment. For the strict conclusion, several additional experiments are required. Here I need to mention the main problem. Authors vary not only the studied factor (salt concentration or pH) but also the protein concentration (less than 100 in acidic conditions and more than 100 in high salt buffer) in experiments. Thus observed effects, in theory, can be explained by the variation in the protein concentration and not by salt or hydrogen ion concentration.
Moreover, the differences in protein concentration AND experimental conditions between experiments are also confusing. For example:
figure 1, 200 or 700 ug/ml TasA in presence of 1.5 M NaCl, but 50 ug/ml TasA at 2.5 pH; 
figure 3, 100 or 800 ug/ml TasA in presence of 1.8 M NaCl, the protein concentration for an acidic condition is not specified.
May be such changes in concentrations do not affect the results, but the question appears. Also, I will advise adding a new section "fibrils preparation" into materials and methods for clarity.

Minor comments.

Why did the authors decide to use formic acid for pH adjusting? In several papers, this substance was used to dissolve amyloid aggregates (for instance Ryzhova et al., 2018. Screening for amyloid proteins in the yeast proteome).

The typos in terms "cross-beta", "alpha-synuclein" (the dash is absent) and "Abeta"(A should be capitalized) should be corrected.

line 194,231,235. The pH value should be added for the description of Tris solutions?

line 69. I would propose to replace "It is antibacterial" with "The protein (or its aggregates) has antibacterial properties".

line 80. "B. subtilis" should be italicized. Also, the title of section 3.3 should be italicized.

line 85. I suppose that "poly (ether sulfone) (PES)" should be replaced with "polyethersulfone (PES)".

ThT concentration is missed in section 2.7.

I would advise adding the legend on fig 3. Also, it will be better to use the same symbols to plot the results of the same experiment. Now turbidity measurements on the main plot and the inset are plotted with points and lines, respectively.

line 168. I would propose to move technical information "Figure 2a shows TasA bundles and figure 2b shows a zoomed-in view to the squared area in figure 2a." to the figure caption.

Panels 2f and 2g are too small. I would advise to increase their size.

Finally I suppose that the value of the paper will be significantly increased if author add experiments showing that aggregates with different morphology have dictint properties in vivo. For example, it is possible to compare the toxicity of the aggregates.

Reviewer 2 Report

In their interesting manuscript, Ghrayeb et al characterized a new path to a different TasA fibril formation, by utilizing a high TasA concentration with a very long incubation for days in the presence of high salt (1.5 M NaCl). In their subsequent comprehensive biophysical characterization of these new kind of TasA fibrils they observed more straight and bundled fibrils, distinct from more tangled fibrils whose formation, induced by a pH shift.

Comments & Questions

-Fig 4 It would be great to show also a CD spectrum from the pH shift induced TasA fibrils to compare those fibrils with the new kind of salt induced straight bundle forming TasA fibrils (As in Fig 1 & 5) .

-Are cross-beta sheets (mentioned in legend of Fig 3) the same as twisted beta-sheets (mentioned in the legends of Fig 4) or are they distinct?

-Discussion p8 e.g. l 291, 295: Why are the authors describing the fibril formation as aggregation?

-Abstract l 18: "secretes" and not "secrets"

- p6 l242 "..we were intrigued to measure.."  sounds not right

Round 2

Reviewer 1 Report

The authors performed great work to improve the manuscript. All my comments were addressed.